# DIRAS3-Derived Peptide Inhibits Autophagy in Ovarian Cancer Cells by Binding to Beclin1

**DOI:** 10.3390/cancers11040557

**Published:** 2019-04-18

**Authors:** Margie N. Sutton, Gilbert Y. Huang, Xiaowen Liang, Rajesh Sharma, Albert S. Reger, Weiqun Mao, Lan Pang, Philip J. Rask, Kwangkook Lee, Joshua P. Gray, Amy M. Hurwitz, Timothy Palzkill, Steven W. Millward, Choel Kim, Zhen Lu, Robert C. Bast

**Affiliations:** 1Department of Experimental Therapeutics, The University of Texas M.D. Anderson Cancer Center, Houston, TX 77030, USA; MNSutton@mdanderson.org (M.N.S.); gilbertyhuang@gmail.com (G.Y.H.); XLiang3@mdanderson.org (X.L.); WMao@mdanderson.org (W.M.); LPang@mdanderson.org (L.P.); pjrask@mdanderson.org (P.J.R.); KLEE74@mgh.harvard.edu (K.L.); 2Department of Pharmacology and Chemical Biology, Baylor College of Medicine, Houston, TX 77030, USA; Rajesh.Sharma@bcm.edu (R.S.); al_reger@yahoo.com (A.S.R.); amymhz@gmail.com (A.M.H.); timothyp@bcm.edu (T.P.); ckim@bcm.edu (C.K.); 3Department of Cancer Systems Imaging, The University of Texas M.D. Anderson Cancer Center, Houston, TX 77030, USA; jpgray@mdanderson.org (J.P.G.); smillward@mdanderson.org (S.W.M.)

**Keywords:** autophagy inhibitor, DIRAS3, Beclin1, peptide therapeutic, ARHI, ovarian cancer

## Abstract

Autophagy can protect cancer cells from acute starvation and enhance resistance to chemotherapy. Previously, we reported that autophagy plays a critical role in the survival of dormant, drug resistant ovarian cancer cells using human xenograft models and correlated the up-regulation of autophagy and DIRAS3 expression in clinical samples obtained during “second look” operations. DIRAS3 is an imprinted tumor suppressor gene that encodes a 26 kD GTPase with homology to RAS that inhibits cancer cell proliferation and motility. Re-expression of DIRAS3 in ovarian cancer xenografts also induces dormancy and autophagy. DIRAS3 can bind to Beclin1 forming the Autophagy Initiation Complex that triggers autophagosome formation. Both the N-terminus of DIRAS3 (residues 15–33) and the switch II region of DIRAS3 (residues 93–107) interact directly with BECN1. We have identified an autophagy-inhibiting peptide based on the switch II region of DIRAS3 linked to Tat peptide that is taken up by ovarian cancer cells, binds Beclin1 and inhibits starvation-induced DIRAS3-mediated autophagy.

## 1. Introduction

Despite progress in surgery and chemotherapy, ovarian cancer, which affects more than 22,000 women in the United States each year, still proves lethal in 70% of cases [1]. Metastatic, drug resistant ovarian cancer cells can remain dormant for years after treatment, only to grow progressively and kill patients. Autophagy, a catabolic process by which long-lived proteins and organelles are degraded into amino acids and fatty acids for catabolism and anabolism, has been shown to sustain dormant cancer cells and enhance resistance to paclitaxel and poly ADP ribose polymerase (PARP) inhibitors [2,3]. Autophagy can facilitate survival of cancer cells in poorly vascularized, nutrient-deprived microenvironments and inhibition of autophagic flux with chloroquine, a functional inhibitor of autophagy, can induce cancer cell death [4]. Treatment of dormant cancer cells with chloroquine can delay their subsequent outgrowth when dormancy is broken [5]. Autophagy can also enhance resistance to paclitaxel, a critical component of primary chemotherapy for ovarian cancer [6,7]. 

We have previously shown that the small GTPase DIRAS3 can induce autophagy and plays an important role in forming the autophagosome initiation complex (AIC) with Beclin1. Autophagy is initiated during starvation when mTOR activity is reduced, leading to the formation of a ULK1/ATG13/FIP200 complex. Further induction of autophagosomes depends upon formation of an autophagy initiation complex (AIC) that contains BECN1 [8,9]. In cells that are not undergoing autophagy, Bcl-2, an anti-apoptotic protein, binds to BECN1/BECN1 dimers and prevents formation of the AIC. During nutrient deprivation, DIRAS3 is induced [10] and facilitates dissociation of BECN1/BECN1 dimers from their negative regulator Bcl-2 [11]. BECN1 dimers are then disrupted and monomers bind DIRAS3 to PIK3C3, a Class III phosphatidylinositol-3-kinase, forming the BECN1/DIRAS3/PIK3C3 AIC [8,12]. The protein ATG14L subsequently associates with and directs the AIC to a phagophore assembly site, activating and initiating the biogenesis of autophagosomes [13,14].

DIRAS3 is an imprinted tumor suppressor gene that is downregulated in 62% of ovarian cancers, as well as many other cancers [15]. DIRAS3 re-expression inhibits cancer cell growth, slows motility, regulates autophagy and sustains tumor dormancy. DIRAS3 induces autophagy through several mechanisms, including downregulation of mTOR, inhibition of FOXO3a phosphorylation, displacing Bcl-2 from BECN1, dissociating BECN1 dimers, and participating directly in the AIC with BECN1 [5,11,16]. DIRAS3 is also required for the induction of autophagy by amino acid starvation and mTOR inhibitor treatment [10]. Within hours of transfer to media lacking amino acids, DIRAS3 is induced at the mRNA and protein level through transcriptional activation by CEBPα and degradation of the transcriptional repressors E2F1 and E2F4 [10]. Upregulation of DIRAS3 induces autophagy consistent with the discovery of increased expression of DIRAS3 and autophagy in small avascular deposits of ovarian cancer found on the peritoneal surface during “second look” operations following primary chemotherapy in patients with no detectable disease by conventional imaging and normal levels of the ovarian cancer biomarker CA125. In addition to inducing autophagy, DIRAS3 can prevent tumor outgrowth and block angiogenesis, inducing dormancy. Clinical observations are consistent with previous laboratory studies using DIRAS3-inducble human ovarian cancer xenografts in nu/nu mice. Induction of DIRAS3 markedly inhibited xenograft growth for several months. However, when DIRAS3 expression was downregulated after 6 to 8 weeks, tumors grew rapidly, at a rate similar to that observed in control mice with low levels of DIRAS3. Interestingly, if autophagy was inhibited functionally by chloroquine treatment while DIRAS3 expression was upregulated, tumors failed to grow upon downregulation of DIRAS3, suggesting a role for autophagy in supporting survival of dormant cancer cells [5]. Thus, we sought to develop a DIRAS3-derived peptide that could inhibit the Beclin1:DIRAS3 interaction necessary for the induction of autophagy, permitting us to test the hypothesis that inhibition of the Beclin1:DIRAS3 interaction would inhibit autophagy required for cancer cell survival and eliminate dormant ovarian tumors that express DIRAS3 and are undergoing autophagy in nutrient poor nodules on the surface of the peritoneum. 

## 2. Results

Given the role of autophagy in facilitating survival of metabolically challenged and nutrient deprived cancer cells, inhibiting autophagy through the specific protein-protein interactions that drive the process could not only provide a novel therapeutic target, but also create a chemical probe to modulate the autophagic process acutely for cell and cancer biology research. The DIRAS3 and BECN1 interaction is essential for the formation of the autophagosome initiation complex (AIC) and for the development of autophagic vesicles, and thus became the focus of our interaction-derived peptide development [11]. To identify regions of DIRAS3 and BECN1 required for AIC formation, we performed peptide array analysis (Figure 1). We generated an array of 15mer or 16mer peptides corresponding to the DIRAS3 (Figure 1B) or BECN1 (Figure 1C) protein sequences, respectively, and then cross probed the arrays with recombinant protein to identify possible interacting sequences. Our peptide array data combined with pull-down experiments (previously published [11]) indicated that, in addition to the N-terminus of DIRAS3 (residues 15–33), the switch II region of DIRAS3 (residues 93–107) interacts directly with BECN1. We hypothesized that DIRAS3 first binds BECN1 through its switch II region, and that inhibiting this interaction with a DIRAS3-derived peptide capable of binding Beclin1, would decrease autophagy and possibly inhibit dormant ovarian cancer cells. 

The DIRAS family proteins (DIRAS1, 2, and 3) share 50%–60% sequence homology, where DIRAS1 and DIRAS2 differ primarily from DIRAS3 in the truncation of the 34-amino acid N-terminus and in the switch II region sequence [17] (Figure 2A). Although the crystal structure of full length DIRAS3 has not been solved, we gained structural insight into the fold of the switch II region of DIRAS3, which is important for the interaction with Beclin1, by developing a DIRAS2/3 chimera. This chimeric protein utilized the previously known crystal structure of DIRAS2 (to anchor the production and analysis) and replaced the switch II sequence of DIRAS2 with that of DIRAS3, permitting us to observe the fold of the DIRAS3 switch II region. Using this new construct, we were able to solve the crystal structure of the DIRAS2/3 chimera (Figure 2B). The DIRAS2/3 protein was crystallized in the hexagonal space group (*P*6_5_22) and diffracted to 3.08 Å resolution. The DIRAS2/3 structure was determined by Phase-MR using poly-Ala model of DIRAS2 crystal structure (PDB ID:2ERX) and refined using Phenix.refine [18,19]. The final model resulted in final R_factor_ of 26.6% and R_free_ of 33.1%. The final model has good geometry with 97.2% residues lying in favored and allowed regions of a Ramachandran plot. The final model includes 1212 non-hydrogen protein atoms equivalent to 164 amino acid residues, 58 non-hydrogen atoms corresponding to ligands (GDP, glycerol, ethylene glycol) and 13 solvent molecules. The root mean square deviations (RMSD) from ideal geometry are 0.002 Å for bond lengths and 0.62 Å for bond angles. Strikingly the structure showed that, despite having a similar fold, the switch II of the DIRAS2/3 chimera has a more extended loop region and shows a positively charged surface with three arginine residues distributed throughout (one at the loop and two at the helix) (Figure 2). Data collection and refinement statistics are shown in Table 1.

Based on the structure of the DIRAS2/3 chimera, we hypothesized that a peptide derived from the switch II region of DIRAS3 might bind BECN1, inhibiting the ability of endogenous full length DIRAS3 to recognize BECN1, thus blocking formation of the autophagy initiation complex. Based on our peptide array and structural data, we designed a candidate peptide (TatD3S2) consisting of the sequence from the switch II region of DIRAS3 (D3S2) conjugated with the HIV-1 Tat protein transduction domain (Tat peptide) to enhance cellular permeability (Figure 3A and Appendix A). A flexible di-glycine (GG) linker was included between the Tat peptide and the D3S2-derived peptide to prevent interaction between the two motifs and add flexibility to the peptide. As a control, a Tat peptide conjugated to a scrambled DIRAS3 switch II sequence (TatD3S2scr) was also synthesized. We first confirmed cellular uptake of the Tat-peptides by conjugating them with fluorescein isothiocyanate (FITC) and performing a dose dependent cellular uptake experiment. A2780 ovarian cancer cells were incubated with FITC-conjugated peptides for 2 h prior to flow cytometry analysis. Increasing concentrations of FITC-Tat-Peptide resulted in FITC cell uptake in a dose dependent manner (Figure 3B). To serve as a control for cellular uptake rather than non-specific membrane binding, we performed a similar experiment where fluorescence of cells was measured after incubation with 50 µM FITC alone, 50 µM FITC-D3S2 peptide without the Tat-leader sequence or 50 µM FITC-Tat-D3S2. Although some uptake of control compounds was observed, addition of the Tat sequence increased uptake by almost four-fold relative to the FITC-labeled D3S2 peptide and almost eight-fold relative to FITC alone (Figure 3C). We confirmed these results using immunofluorescence staining of A2780 ovarian cancer cells treated for 2 h with FITC, FITC-D3S2 and FITC-Tat-D3S2 (Figure 3D). 

Binding of the D3S2 peptide to Beclin1 was established using a Beclin1 peptide array (16mers stepped by three amino acids). Using biotinylated-D3S2, we determined that the peptide showed significant activity with amino acids 250–277 (Figure 4A) within the coil-coiled domain of Beclin1 overlapping with the predicted interacting domains of full-length DIRAS3 (Figure 1C). Using FlexPepDock server [20,21], we docked the modeled peptide to Beclin1 crystal structure (PDB ID:4DDP) (Figure 4B). The interacting residues of Beclin1 and docked peptide (zoom-in view) are shown in sticks and the polar interactions (five hydrogen bonds) between them reveals a strong binding. The binding analysis is generally consistent with the peptide array data and suggests additional modifications of D3S2 that may enhance Beclin1 affinity and/or selectivity. We further confirmed binding between the peptide and Beclin1 using surface plasmon resonance (SPR) [22]. Compared to human serum albumin (HSA), the interactions between the control peptides and the DIRAS3 switch II-derived peptide showed specificity for Beclin1 where interactions could be detected from 5 to 405 nM concentrations of the protein and little to no binding was observed with HSA at 1–16 µM concentrations (Figure 4C). From this analysis we determined that the Tat-leader sequence provided some non-specific binding to Beclin1, so we proceeded using only the D3S2 peptide biotinylated at the C-terminus to determine the binding affinity by SPR. Using concentrations of Beclin1 ranging from 10 to 270 nM, we determined that the DIRAS3 switch II-derived peptide (D3S2) has an equilibrium dissociation constant (*K*_D_) of 1.90 (±0.20) nM with full-length Beclin1 (Figure 4D). The association rate constant (*k_a_*) is 6.22 (±0.56) × 10^4^ and dissociation rate constant (*k*_d_) is 1.25 (±0.04) × 10^−4^. 

To confirm the functionality of Tat-D3S2, we tested its effects on A2780 and SKOv3 cells in which autophagy was induced by amino acid deprivation. Using CellTiter-Glo, a luminescent cell viability assay that quantifies the amount of ATP present as an indicator of metabolically active cells, we determined cell viability following peptide treatment with or without amino acid deprivation. We found that upon amino acid deprivation, the addition of the DIRAS3 switch II-derived peptide (Tat-D3S2) and the control peptide significantly inhibited cell viability compared to the Tat-GG or no peptide controls (Figure 5A,B). The inhibition of cell viability was correlated to the binding affinity of the peptides to Beclin1, and showed specificity to those cells undergoing autophagy as the cells cultured with the peptides in complete media did not show significant growth inhibition. To determine if the effect on cell viability was specifically due to blocking DIRAS3-mediated autophagy upon amino acid deprivation, both cell lines were pre-treated with 50 µM peptides Tat-GG and Tat-Scr serving as controls, and Tat-D3S2 for 2 h prior to changing the media to one lacking amino acids and re-treating the cells for 2 h. Cell lysates were collected, and Western blot analysis was performed documenting a significant induction in autophagy following amino acid deprivation in untreated cells, and those treated with 50 µM of Tat-GG or Tat-Scr peptide. Tat-D3S2 treatment decreased the induction of autophagy for both cell lines as determined in a decreased LC3I/LC3II conversion ratio (Figure 5C). Transmission electron microscopy was used to confirm these results and documented a decrease in double membrane autophagosomes following amino acid deprivation in SKOv3 cells treated with the autophagy inhibiting DIRAS3 switch II-derived peptide, confirming the Western blot analysis (Figure 5D,E). 

## 3. Discussion

Development of a DIRAS3 switch II-derived peptide that binds Beclin1 and inhibits DIRAS3-mediated autophagy under nutrient deprivation represents a novel approach to inhibiting autophagy by blocking a specific protein-protein interaction (PPI). Our studies suggest that the cell permeable Tat-D3S2 derived peptide takes advantage of the direct similarity to the DIRAS3 protein fragment which engages with Beclin1 to form the AIC. In contrast to autophagy inhibitors in clinical development that functionally inhibit fully developed autophagosomes (e.g., chloroquine, hydroxychloroquine, chloroquine dimers and quinacrine dimers) this approach inhibits autophagy through the selective disruption of a PPI critical for autophagosome initiation. This first generation peptide inhibitor has demonstrated feasibility of targeting a specific PPI critical to the induction of autophagy.

While peptides remain the most widely studied medium-sized (1–2 kDa) inhibitors of PPIs, their clinical efficacy has been limited based on the low proteolytic and conformational stability [23,24], both of which may play a critical role in reducing the efficacy of this first generation peptide. From our study, we were able to determine that the DIRAS3 switch II-derived peptide bound tightly to Beclin1 (*K*_D_ ~1.9 nM), however high doses (50 µM) were used in vitro to observe a robust effect on amino acid starvation-induced autophagy. Interestingly, we also observed that the current control peptide (Tat-Scr) showed some affinity for Beclin1 and this correlated with the effect on cell viability (Figure 5A,B). Although these concentrations would likely not be achievable in vivo, future studies to characterize the thermodynamics and kinetics of the Tat-D3S2 peptide and the DIRAS3:Beclin1 interaction that it targets, may further inform the design of more potent, stable variants that can overcome some of the challenges currently faced by our DIRAS3 switch II-derived autophagy inhibiting peptide.

Aside from the role in targeting autophagy to eliminate dormant tumors, identification of a selective autophagy inhibitor could provide therapeutic efficacy for other diseases which are defined by excessive autophagy including Diamond-Blackfan anemia, heart disease or neurodegenerative diseases that show excessive autophagosome accumulation [25,26]. This opens the potential scope of therapeutic efficacy of autophagy inhibiting therapies.

## 4. Materials and Methods 

### 4.1. Antibodies and Reagents

Antibodies against LC3 (2775S) and β-actin (4910L), were purchased from Cell Signaling Technology (Danvers, MA, USA). P62/SQSTM1 antibody was purchased from MBL (Woburn, MA, USA). 

### 4.2. Cell Lines

Human ovarian cancer cells, A2780 and SKOv3 were grown in RPMI, supplemented with 10% fetal bovine serum and 1% L-glutamine. Amino acid free media was prepared from RPMI powder (US Biological R8999-04A, Salem, MA, USA) supplemented with 10% dialyzed (MWCO 10k) fetal bovine serum (26400-044).

### 4.3. Peptides

High purity (>95%) peptides were obtained from GenScript (Piscataway, NJ, USA). During peptide synthesis, peptides were labeled with N-terminal FITC or Biotin for additional studies. Freeze-dried peptides were reconstituted in high-purity water.

### 4.4. Brightfield and Fluorescence Microscopy

Microscopy was performed using an Olympus IX71 microscope equipped with a DP72 camera and an XM10 camera at the specified magnifications.

### 4.5. Transmission Electron Microscopy

Cells were seeded at (1.0–2.5 × 10^5^ cells/well) in a 6-well plate and incubated for different intervals in amino acid-free medium. Samples were then fixed with light Karnovsky’s fixative solution containing 3% glutaraldehyde plus 2% paraformaldehyde in 0.1 M cacodylate buffer at pH 7.3 and stored at 4 °C. After fixation, samples were submitted to the MD Anderson electron microscopy core facility for processing (Mr. Kenneth Dunner Jr.). Briefly, cells were washed in 0.1 M cacodylate buffer and treated with 0.1% Millipore-filtered buffered tannic acid, postfixed with 1% buffered osmium tetroxide for 30 min, and stained with 1% Millipore-filtered uranyl acetate. The samples were washed several times in water, then dehydrated in increasing concentrations of ethanol, infiltrated, and embedded in LX-112 medium. The samples were polymerized in a 60 °C oven for 2 days. Ultrathin sections were obtained using a Leica Ultracut microtome (Leica), stained with uranyl acetate and lead citrate in a Leica EM Stainer, and examined in a JEM 1010 transmission electron microscope (JEOL Inc., Peabody, MA, USA,) at an accelerating voltage of 80 kV. Digital images were obtained using AMT Imaging System (Advanced Microscopy Techniques Corp, Woburn, MA, USA). Quantification was performed manually, counting the average number of autophagosomes (*n* >5 for each condition).

### 4.6. Western Blotting

Cell lysates were prepared as indicated following incubation in lysis buffer (50 mM Hepes, pH 7.0, 150 mM NaCl, 1.5 mM MgCl_2_, 1 mM EGTA, 10 mM NaF, 10 mM sodium pyrophosphate, 10% glycerol, 1% Triton X-100) plus protease and phosphatase inhibitors (1 mM PMSF, 10 µg/mL leupeptin, 10 µg/mL aprotinin and 1 mM Na_3_VO_4_). Cells were lysed for 30 min on ice, and then centrifuged at 17,000 × *g* for 30 min at 4 °C. The protein concentration was assessed using a bicinchoninic acid (BCA) protein assay (ThermoScientific, Waltham, MA, USA, #23225). Equal amounts of protein were separated by 8%–16% SDS-PAGE, transferred to PVDF membranes and subjected to Western blotting using an ECL chemiluminescence reagent (PerkinElmer, Hopkinton, MA, USA, #NEL105001).

### 4.7. Peptide Array Analysis

Peptide arrays were made using the MultiPep RS robot (Intavis, Bergisch Gladbach, Germany) according to the SPOT synthesis technique described by Frank et al. (2002) [27]. Arrays were developed by soaking membranes in 100% methanol for 10 min at room temperature followed by washes with PBS (3 × 10 min). Membranes were then blocked overnight at 4 °C in 5% BSA/PBS. Recombinant proteins were added to the membrane at a final concentration of 1 µg/mL in 1% BSA/PBS and shaken gently at room temperature for 2 h. Membranes were washed three times with 1% BSA/PBS for 10 min each prior to the addition of primary antibody diluted in wash buffer for 1 h at room temperature. The membrane was washed three times for 10 min and a diluted secondary antibody (1:10,000) was added for 45 min at room temperature, with gentle shaking. The membrane was washed three times with wash buffer for 10 min, then followed with three washes with PBS-T (PBS containing 0.1% Tween-20) for 10 min each. Membranes were developed with HRP substrates and exposed to X-ray film. 

### 4.8. Protein Expression and Purification

A DIRAS2/3 chimera construct was generated by replacing the amino acid residues 92–108 of DIRAS3 with 62–78 amino acid residues of DIRAS2. The DIRAS2/3 chimera construct was cloned into pQTEV vector using ligation-independent cloning and transformed into BL21 (DE3) *Escherichia coli* cells. Cells carrying the DIRAS2/3 chimera constructs were grown at 37 °C until the OD at 600 nm (OD_600_) reached to 0.6 and protein expression was induced with 0.5 mM isopropyl β-D-1-thiogalactopyranoside. Cultures were grown for an additional 12 h at 18 °C and harvested by centrifugation. Cells were resuspended in lysis buffer containing 50 mM Tris (pH 7.5), 150 mM sodium chloride, 10 mM magnesium chloride, 10 mM Imidazole, 5% glycerol and 1 mM β-mercaptoethanol and lysed using a cell disrupter (Constant Systems, Northamptonshire, UK). The lysate was cleared by centrifugation at 35,000 × *g* for 2 h at 4 °C. Cleared lysate was passed through a 0.22-μm filter. The recombinant DIRAS2/3 protein was purified by immobilized metal ion affinity chromatography using a Profinia system (Bio-Rad) and protein was eluted with resuspension buffer supplemented with 250 mM imidazole. The eluted fractions were pooled and incubated with tobacco etch virus (TEV) protease to cleave the His tag. Both the protease and cleaved His-tags were removed by reloading the dialyzed samples onto the Ni-NTA column. Buffer exchange and aggregate removal were performed using FPLC size-exclusion chromatography on a Hi-Load 16/60 Superdex-75 column (GE Healthcare) in FPLC buffer containing 50 mM Tris (pH 7.5), 150 mM sodium chloride, 10 mM magnesium chloride and 1 mM TCEP-HCl. The purified recombinant DIRAS2/3 protein was concentrated to 30 mg/mL for crystallization. Beclin1 recombinant protein was produced in a similar fashion. 

### 4.9. Crystallization, Data Collection, Phasing, Model Building and Refinement

For the crystallization of the DIRAS2/3 chimera, the protein was concentrated to 30 mg/mL supplemented with 5 mM GTP. Crystals appeared in 1500 mM Ammonium sulfate, 100 mM Tris base/Hydrochloric acid pH 8.5, and 12% glycerol. The crystals were harvested and immersed in 20% ethylene glycol, immediately before being flash-cooled in liquid nitrogen. The crystals were then diffracted at the Advanced Light Source (ALS) at Berkeley, CA, USA. The crystal structure of DIRAS2/3 was determined by molecular replacement (MR) method (Phenix.phaser) using the crystal structure of DIRAS2 (PDB ID:2ERX) as search model. Refinement and model building were performed using Phenix.refine and COOT programs [18,19,28]. The PyMol visualization program was used for structural analysis of the final refined model and drawing of figures. The validation of final refined model was done using the MOLPROBIDITY server [29].

The data collection and refinement statistics are summarized in Table 1.

### 4.10. Surface Plasmon Resonance

SPR experiments were performed using a Biacore 3000 optical biosensor (GE Healthcare/Biacore AB) at 25 °C. CM3 sensor chip, sensor chip SA (Carboxymethylated dextran pre-immobilized with streptavidin) and amine coupling kit (1-ethyl-3-(3-dimethylaminopropyl) carbodiimide hydrochloride, N-hydroxysuccinimide, 1.0 M ethanolamine-HCl pH 8.5) were purchased from GE Healthcare. Human serum albumin was purchased from Sigma (A3782) (Sigma, St. Louis, MO, USA). Immobilization of peptides (all have the same pI of 12.3) was conducted at a flow rate of 5 µL/min in PBS (8.06 mM Na_2_HPO_4_ and 1.94 mM KH_2_PO_4_, 2.7 mM KCl, 137 mM NaCl, pH 7.4). Using the amine coupling kit, the chip surface was activated for 4 min followed by 4 min injection of peptide solution in PBS (35 µg/mL for Tat-D3S2 and Tat-Scr, and 20 µg/mL for smaller Tat-GG, respectively), and then deactivated for 4 min. A flow cell with the same treatment but no peptide coupled was used as a reference surface. Binding experiments were performed at a flow rate of 30 µL/minute in 20 mM Tris (pH8.0), 200 mM NaCl, 0.02% Tween 20, 5% glycerol. For affinity measurement, c-terminal biotinylated D3S2 peptide (10 nM) was captured on a sensor chip SA at a flow rate of 10 µL/min in 25 mM Tris (pH 7.5), 150 mM NaCl, 0.02% Tween 20. Regeneration of the peptide surface was achieved by short injection of NaCl/NaOH (1 M/5 mM) solution followed by 0.01% SDS. Reference and buffer corrected SPR response was collected and the data was analyzed using Biacore evaluation software. The kinetic parameters and *K*_D_ value were determined by fitting the sensorgram to a Langmuir 1:1 binding model (BIAevaluation version 4.1.1, GE Healthcare/Biacore AB, Marlborough, MA, USA).

### 4.11. Analysis of Growth Inhibition Following Peptide Treatment

SKOV3 (1200 cells/well) and A2780 (2000 cells/well) were seeded in 96-well plates and allowed to adhere. Then, 24 h post seeding, cells were treated with peptides at the indicated concentration and allowed to incubate at 37 °C for 56 h (A2780) or 72 h (SKOV3) in the presence of full media or media lacking amino acids, as previously described. Upon completion of incubation, CellTiter-Glo (Promega G7570) was added to the plate and incubated for 10 min prior to reading the luminescence on a Synergy5 Biotek plate reader. 

### 4.12. Analysis of Peptide Internalization

A2780 cells were seeded in 100 mm dishes at 1.5 × 10^6^ cells/plate 24 h prior to incubation with the peptides. FITC labeled peptides (ThermoScientific 53027) were added to the cells at the specified concentrations and cells were incubated at 37 °C for 2 h. Following incubation, cells were washed three times with PBS for 5 min each prior to trypsinization and resuspension in PBS. Fluorescence-activated cell sorting (FACS) analysis was performed using FACSJazz.

## 5. Conclusions

We report the development of an autophagy inhibiting peptide which was designed to antagonize the Beclin1:DIRAS3 interaction of the autophagosome initiation complex (AIC). Through X-ray crystallography and peptide array analysis we modeled the switch-II region of DIRAS3 and derived the inhibitory peptide, capable of binding Beclin1 from the DIRAS3 sequence. Modifications to the peptide enhanced cell permeability, specifically the addition of the HIV Tat peptide leader sequence followed by a di-glycine linker. We document uptake of the peptide to intact ovarian cancer cells in a dose dependent manner. Induction of autophagy by nutrient deprivation (removal of amino acids from the culture media) can be inhibited by the DIRAS3 switch II-derived peptide but not by the leader sequence peptide (Tat-GG) or a scrambled control peptide. Taken together, these data demonstrate that autophagy inhibition by Tat-GG-D3S2 peptide can be achieved through inhibition of the functional protein-protein interaction (PPI) of DIRAS3 and Beclin1. 

## Figures and Tables

**Figure 1 cancers-11-00557-f001:**
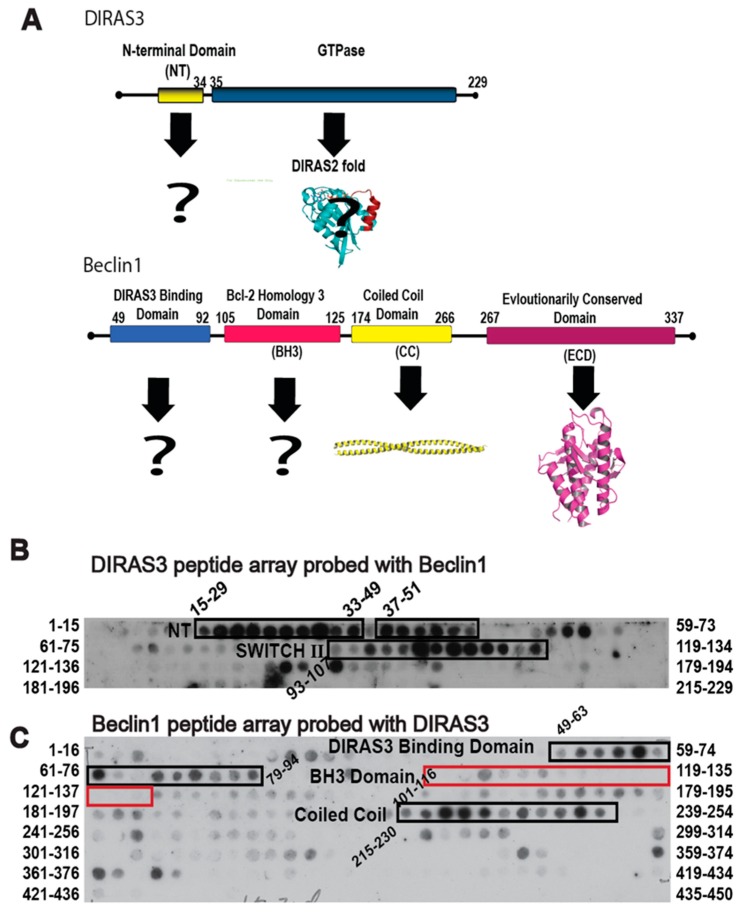
DIRAS3 and BECN1 domain organization and peptide overlay. (**A**) Functional domains are shown with their boundaries marked with residue numbers. Crystal structures of the CC and ECD domains are shown. (**B**) Beclin1 overlay of a peptide array comprising residues 1–229 of DIRAS3. (**C**) DIRAS3 overlay of a peptide array comprising residues 1–450 of Beclin1. A 16mer peptide array staggering every other residue was probed with DIRAS3. The BH3 domain (105–125, red box) of Beclin1, which binds Bcl-2, does not bind DIRAS3.

**Figure 2 cancers-11-00557-f002:**
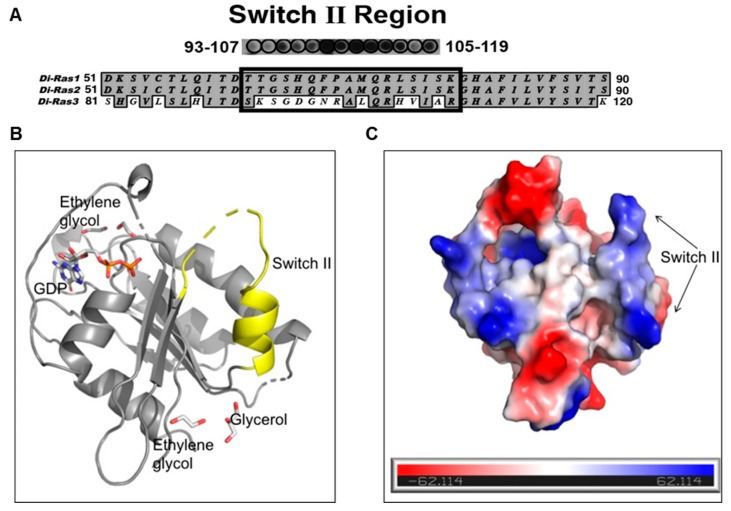
DIRAS2/3 chimera identifies structural confirmation of the switch II region of DIRAS3. (**A**) Peptide array data and sequence alignment of DIRAS of the switch II region responsible for interaction with the N-term of Beclin1. (**B**) Based on the sequence alignment and peptide array data of DIRAS3, we engineered a chimera (on the right) using the crystal structure of DIRAS2 (PDB code:2ERX) as a template. The crystal structure of the chimera is shown in the left panel. The switch II regions are colored in yellow. (**C**) The surface electrostatic potential of DIRAS2/3 chimera protein.

**Figure 3 cancers-11-00557-f003:**
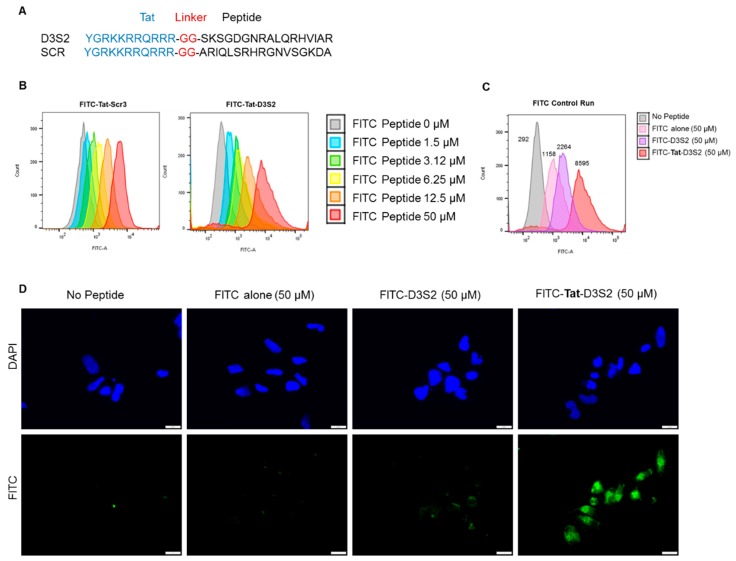
Tat-conjugated DIRAS3 switch II-derived peptide enters ovarian cancer cells in a dose dependent manner. (**A**) Tat-conjugated sequence of the DIRAS3 switch II-derived peptide and its scrambled control. (**B,C**) Flow cytometry quantification of peptide uptake by A2780 ovarian cancer cells incubated with the peptide for 2 h at 37 °C with increasing concentrations of fluorescein isothiocyanate (FITC)-labeled D3S2 peptide. (**D**) Immunofluorescence staining of A2780 ovarian cancer cells following treatment with FITC alone, FITC-D3S2 and FITC-Tat-D3S2. Scale bars indicate 20 nm.

**Figure 4 cancers-11-00557-f004:**
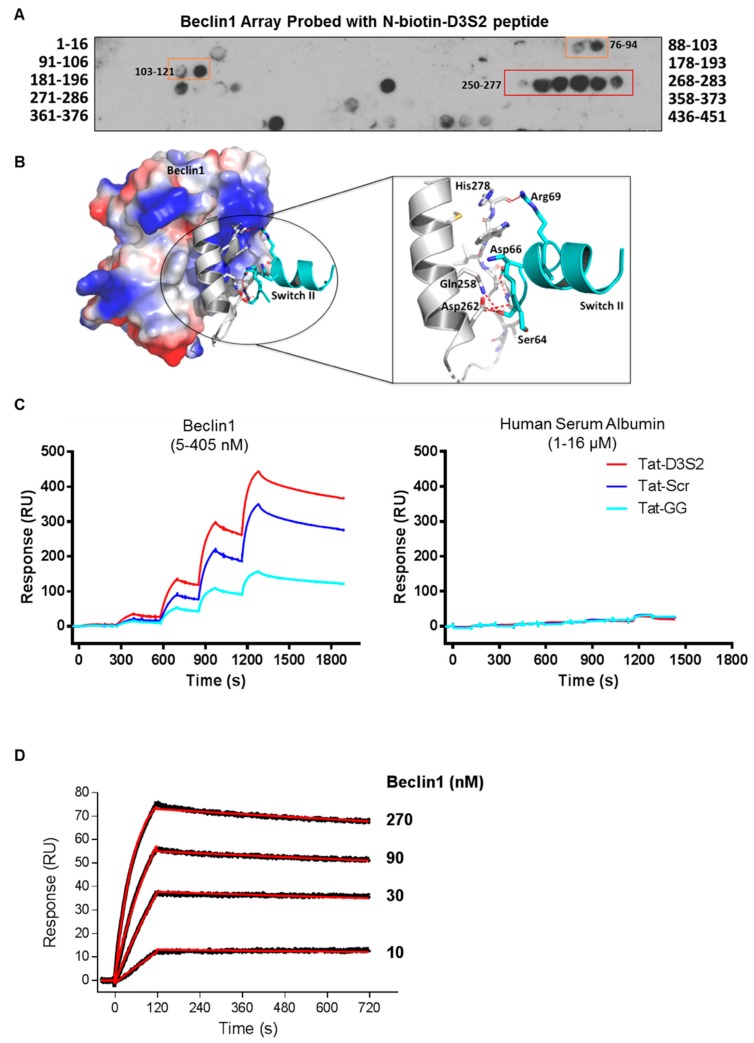
DIRAS3 switch II-derived peptide interacts with Beclin1. (**A**) N-biotin-D3S2 peptide overlay of a peptide array comprising residues 1−450 of Beclin1. A 16mer peptide array staggering every other residue was probed with the DIRAS3 switch II-derived peptide. Interaction between the peptide and amino acids 250−277 (red box) of Beclin1 was revealed. (**B**) FlexPepDock server was used for docking the modeled peptide to Beclin1 crystal structure (PDB ID:4DDP). Beclin1 is depicted in electrostatic surface, while the probable peptide binding region of Beclin1 and docked peptide are shown in grey and cyan colors, respectively. (**C**) Surface plasmon resonance (SPR) was used to determine the binding specificity of the DIRAS3 switch II-derived peptides with Beclin1. SPR was performed with Tat-GG (~150 RU), Tat-Scr (~280 RU) or Tat-D3S2 (~280 RU) peptides immobilized in CM3 sensor chip (GE). Recombinant Beclin1 (5, 15, 45, 135 and 405 nM) or human serum albumin (1, 2, 4, 8 and 16 µM) protein were titrated sequentially onto the peptide sensor and the sensograms obtained. (**D**) To determine the binding affinity of the DIRAS3-derived switch II peptide (D3S2) to Beclin1, c-terminal biotinylated D3S2 peptide (~120 RU) was immobilized and Beclin1 recombinant protein was injected at the concentrations as indicated. The sensograms (shown in black) were obtained and the association rate (*k*_a_), dissociation rate (*k*_d_) and equilibrium dissociation (*K*_D_) constants were determined from fitting (shown in red).

**Figure 5 cancers-11-00557-f005:**
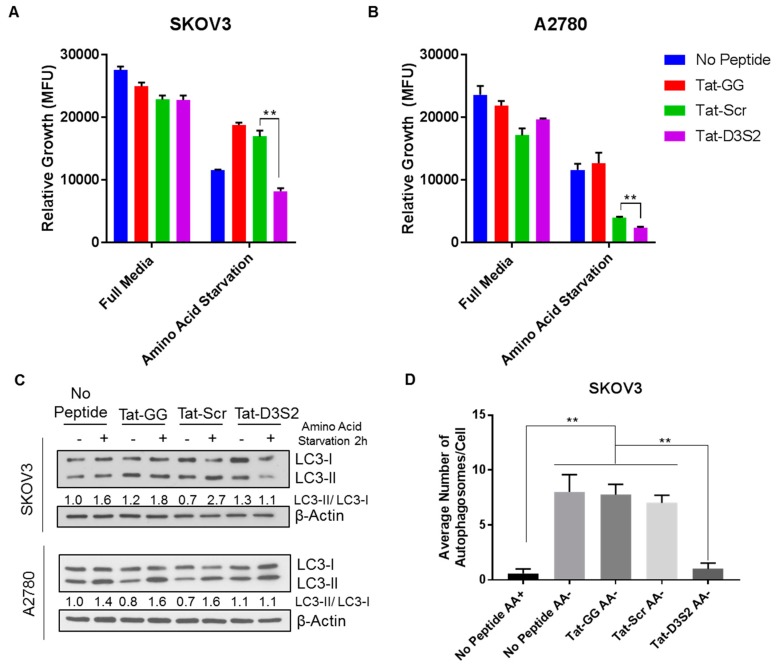
DIRAS3 switch II-derived peptide inhibits amino acid deprivation-induced autophagy in ovarian cancer cells. (**A**) SKOV3 and (**B**) A2780 ovarian cancer cells were seeded in 96-well plates at 1200–2000 cells/well, respectively and treated with indicated peptides (100–200 µM, respectively). The cells were incubated with peptides in full media or media lacking amino acids for 52–76 h, and cell viability was measured by luminescence. The experiment was performed in triplicate and significance denoted by ** *p* < 0.001. (**C**) SKOV3 and A2780 ovarian cancer cell lines were pre-incubated with the peptide treatments (50 µM) as indicated for 2 h prior to undergoing amino acid deprivation for an additional 2 h. Autophagy induction was determined by the ratio of LC3 II to LC3 I. Treatment with the DIRAS3-derived switch II peptide (Tat-D3S2) significantly inhibited autophagy induction compared to treatment with the control peptides (Tat-GG and Tat-Scr). (**D**) Quantification of the average number of autophagosomes per cell (** *p* < 0.01) was determined from transmission electron microscopy images performed of SKOV3 ovarian cancer cells following treatment with the peptides and amino acid deprivation as described previously. (**E**) Autophagosomes are represented by double membrane vesicles and denoted with red arrows.

**Table 1 cancers-11-00557-t001:** Refinement statistics for DIRAS2/3 Chimera X-ray crystallography.

Refinement Statistic	DIRAS 2/3 Chimera
Resolution range	58.62 to –3.081 (3.192–3.081)
Space group	P 6_5_ 2 2
Unit cell (Å)	a = 92.03 b = 92.03 c = 86.5202
Total reflections	
Unique reflections	4323 (406)
Multiplicity	23.5 (24.0)
Completeness (%)	99.98 (100.00)
Mean I/sigma(I)	13.2 (2.6)
Wilson B-factor	58.67
R-merge	0.30 (1.47)
CC1/2	0.99 (0.76)
Reflections used in refinement	4323 (406)
R-work	0.2660 (0.2261)
R-free	0.3310 (0.2706)
Number of non-hydrogen atoms	1212
macromolecules	1141
ligands	58
solvent	13
Protein residues	164
RMS (bonds)	0.002
RMS (angles)	0.62
Ramachandran favored (%)	88.59
Ramachandran allowed (%)	8.72
Ramachandran outliers (%)	2.68
Clashscore	5.52
Average B-factor	51.27
macromolecules	51.44
ligands	49.17
solvent	46.11

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
