# Peer review of "DIRAS3-Derived Peptide Inhibits Autophagy in Ovarian Cancer Cells by Binding to Beclin1"

_cancers, 2019, doi:10.3390/cancers11040557_

Round 1

Reviewer 1 Report

I think that the Authors have addressed in a satisfactory way to the criticims raised by the reviewers.

I am personally in favor of acceptance.

Author Response

No comment

Reviewer 2 Report

The revised manuscript can be accetped for publishing in Cancers now.

Author Response

No comment

Reviewer 3 Report

The manuscript is improved and clarified with the revised cover letter rebuttal.

I still have some concern about the concentration that inhibits autophagy versus indirect affect on viability. The authors say they dosed the peptide and found viability was compromised at approximately 25 uM. However, in Figure 5 the authors indicate the used a concentration of 100-200 uM.  If I understand correctly, viability will be impacted based on their dose response curve.  Is this correct for the assays in Figure 5?  Please clarify.

Given that the concentrations used in vitro may not be possible in vivo, I strongly advise that line 269-270 be removed and some text be added to convey that new iterations of the peptide formulation need to be developed for any possible therapeutic potential.  I don't want the paper to give the impression that this version of the peptide can go directly in vivo into mouse models.

Author Response

I still have some concern about the concentration that inhibits autophagy versus indirect affect on viability. The authors say they dosed the peptide and found viability was compromised at approximately 25 uM. However, in Figure 5 the authors indicate the used a concentration of 100-200 uM.  If I understand correctly, viability will be impacted based on their dose response curve.  Is this correct for the assays in Figure 5?  Please clarify.

The cell viability assays were performed at higher concentrations (100-200 uM)  over 56-72 hours, whereas the autophagy studies were performed at lower concentrations and on a shorter timescale (50 uM for 2-4 hrs). Our cell based functional assays were performed such that we would not be altering cell viability (50 uM concentration and the shortened time course) allowing us to adequately attribute the role of the peptides on autophagy to that treatment rather than off target toxicities associated with cell death.

Given that the concentrations used in vitro may not be possible in vivo, I strongly advise that line 269-270 be removed and some text be added to convey that new iterations of the peptide formulation need to be developed for any possible therapeutic potential.  I don't want the paper to give the impression that this version of the peptide can go directly in vivo into mouse models.

We have removed line 269-270 and updated the discussion to clarify this point.

Round 2

Reviewer 3 Report

Thank you for the clarifications. All of my issues have been addressed.

This manuscript is a resubmission of an earlier submission. The following is a list of the peer review reports and author responses from that submission.

Round 1

Reviewer 1 Report

From an array of 15-16 mer peptides, the authors have identified a peptide able to disrupt the DIRAS3-BECLIN1 interaction, thus inhibiting the formation of the autophagy initiation complex. The inhibitory peptide conjugated to a TAT sequence was proven to enter ovarian cancer cells and to prevent starvation-induced autophagy. Authors have chosen the LC3I to LC3II conversion and electron microscopy of vacuoles as endpoints to demonstrate the effect of the peptide on autophagy. This is a nice piece of work, well designed and  well performed. Controls have been included. The results support the conclusion. Though not mandatory, the authors could have gone further showing the inhibition of Beclin1-Vps34 interaction (by co-ipp) or the lack of PIP3 formation (e.g., co.trasfection with GFP-FYVE). Minor revision is required to fix some typos and grammar/syntax errors. Here, I report a few: 

line 44, delete full stop after "and"; line 49, delete comm after DIRAS3; line 77, delete "were"; line 81, delete "water", line 144, delete "Where". 

Author Response

Reviewer #1:

Comments and Suggestions for Authors

From an array of 15-16 mer peptides, the authors have identified a peptide able to disrupt the DIRAS3-BECLIN1 interaction, thus inhibiting the formation of the autophagy initiation complex. The inhibitory peptide conjugated to a TAT sequence was proven to enter ovarian cancer cells and to prevent starvation-induced autophagy. Authors have chosen the LC3I to LC3II conversion and electron microscopy of vacuoles as endpoints to demonstrate the effect of the peptide on autophagy. This is a nice piece of work, well designed and well performed. Controls have been included. The results support the conclusion. Though not mandatory, the authors could have gone further showing the inhibition of Beclin1-Vps34 interaction (by co-ipp) or the lack of PIP3 formation (e.g., co.trasfection with GFP-FYVE). Minor revision is required to fix some typos and grammar/syntax errors. Here, I report a few: 

line 44, delete full stop after "and"; line 49, delete comm after DIRAS3; line 77, delete "were"; line 81, delete "water", line 144, delete "Where". 

We have made the suggested grammatical/syntax corrections and have performed additional proofreading to eliminate additional typos and grammatical errors.

Reviewer 2 Report

Comments:

Comments

1.    In the introduction section, the rational and hypothesis of this study must be described precisely.

2.    In Fig. 2, the amino acid sequence between DIRAS2 and DIRAS3 just shares 50-60% homology and their present the specific different on the switch II region. Base on these backgrounds, does the crystal structure of DIRAS2 really suitable as a template to capitalize on the crystal structure of DIRAS3?   

3.    As mention in the line-178, …higher p62 expression (Fig. 4C-D). However, there are no any data associated with p62 protein were showed.

4.    Please show the results of the cellular biological function assay, such as cell survival rate, by the effect of Tag-D3S2 and Beclin 1 interaction.

5.    The discussion section is too rough.

Author Response

Comments

1.    In the introduction section, the rational and hypothesis of this study must be described precisely.

We have edited the text of the introduction to include the rationale and hypotheses that we aimed to test with this study (Lines 86-91).

2.    In Fig. 2, the amino acid sequence between DIRAS2 and DIRAS3 just shares 50-60% homology and their present the specific different on the switch II region. Base on these backgrounds, does the crystal structure of DIRAS2 really suitable as a template to capitalize on the crystal structure of DIRAS3?  

Previous work from our group has sought to crystalize full-length DIRAS3, but due to the complexity of the protein and flexibility of the N-terminus, this has remained a challenge. Based on these attempts and the known crystal structures of DIRAS1 and DIRAS2, we decided to gain structural insight of the switch II region of DIRAS3, by developing a chimera protein where the switch II region of DIRAS2 was replaced with that of DIRAS3, testing the hypothesis that by only swapping the switch II region we would be able to obtain a more favorably acting protein and therefore the crystal structure could be solved allowing for further investigation of the switch II domain of DIRAS3. Based on this comment, we have edited the text to make this approach more clear and better explain our intent of developing the chimera protein (Lines 115-123).

3.    As mention in the line-178, …higher p62 expression (Fig. 4C-D). However, there are no any data associated with p62 protein were showed.

      We have removed this typo from the text (Now, Line 223).

4.    Please show the results of the cellular biological function assay, such as cell survival rate, by the effect of Tag-D3S2 and Beclin 1 interaction.

We have added cell viability survival analysis with or without the addition of amino acids to the media and included this as Figure 5A-B.

5.    The discussion section is too rough.

We have made changes to the discussion section which has hopefully improved the flow.

Reviewer 3 Report

Brief Synopsis

This group previously identified DIRAS3 as an important regulator of starvation-induced autophagy in ovarian cancer cells. Here they follow-up the story by identifying a peptide that binds to beclin1 and disrupts its interaction with DIRAS3. As a result of this “DN-peptide” interaction, amino acid-induced autophagy activity is blocked.

Major Comments

Overall this is a convincing paper using structure-function analysis and chemistry to identify a potential druggable target to block autophagy.  Although the data largely support the conclusions of the paper, some problems need to be addressed.

There are no functional readouts of blocking autophagy with this peptide nor is there careful evaluation of kinetics or dose.  How durable is the peptide in blocking autophagy? Does the peptide kill cells? If so, at what dose range?  Will it preferentially kill ovarian cancer ‘stem-like’ cells such as side-population or ALDH1+ SKOV3 or A2780 cells?

Can the authors confirm that peptide-treated cells have disrupted beclin1-DIRAS3 interaction (e.g. IP)?

It is somewhat surprising that the authors did not go the extra-step of showing the peptide is blocking autophagy in vivo. Even a small pilot experiment would really substantiate their findings that this peptide could be translatable to the clinic.

Minor

The claim that autophagy is inhibited by the peptide should be substantiated with a controlled and appropriate autophagy flux assay (Fig. 4C-D), and quantified EM images for reduction in double-membrane autophagosomes.

Author Response

Major Comments

Overall this is a convincing paper using structure-function analysis and chemistry to identify a potential druggable target to block autophagy.  Although the data largely support the conclusions of the paper, some problems need to be addressed.

1.    There are no functional readouts of blocking autophagy with this peptide nor is there careful evaluation of kinetics or dose.  How durable is the peptide in blocking autophagy? Does the peptide kill cells? If so, at what dose range?  Will it preferentially kill ovarian cancer ‘stem-like’ cells such as side-population or ALDH1+ SKOV3 or A2780 cells?

To address this concern, we have performed cell viability assays with or without the addition of amino acids to the culture media (presented in Figure 5A-B). While we do not know whether the effect of the peptide will preferentially kill ovarian cancer ‘stem-like’ cells, we only observed peptide-induced cell killing in those cases in which the cells were undergoing autophagy, as induced by amino acid deprivation.

Dose response curves document inhibited cell viability starting around 25 µM.

2.    Can the authors confirm that peptide-treated cells have disrupted beclin1-DIRAS3 interaction (e.g. IP)?

Although amino acid deprivation upregulates DIRAS3 expression in ovarian cancer cells, it is still difficult to detect the protein expression by western blot analysis therefore making it difficult to detect the DIRAS3:Beclin1 interaction at physiological concentrations without overexpressing the proteins. For this reason we performed SPR analysis to document the binding affinity between the Tat-D3S2 peptide and Beclin1 compared to the controls Tat-GG and Tat-Scr. The SPR analysis documented a tight binding of the peptide to the protein target (Beclin1) with a KD ~1.9 nM. This data has been included in Figure 4C-D.

3.    It is somewhat surprising that the authors did not go the extra-step of showing the peptide is blocking autophagy in vivo. Even a small pilot experiment would really substantiate their findings that this peptide could be translatable to the clinic.

While we agree with the reviewer that an in vivo animal study would facilitate translation of this autophagy-inhibiting peptide to the clinic, we believe that the current concentration needed to achieve a robust effect on blocking DIRAS3-mediated autophagy would limit its efficacy, as these concentrations are likely not achievable in vivo. Future studies to enhance the stability of the peptide and increase potency are planned with the goal of developing a lead candidate peptide that can be further pursued as a translatable autophagy inhibitor.

Minor

1.    The claim that autophagy is inhibited by the peptide should be substantiated with a controlled and appropriate autophagy flux assay (Fig. 4C-D), and quantified EM images for reduction in double-membrane autophagosomes.

We have quantified the TEM images and included that as Figure 5D.